# Learning Multi-Agent Communication using Regularized Attention Messages

## Abstract

Learning how to communicate in Multi-Agent Reinforcement Learning (MARL) can be key to solve complex cooperative tasks. Recent approaches have shown the advantages of using an efficient communication architecture, tackling problems such as what, when, or whom to communicate. However, these methods still fail to solve some complex scenarios, and some of them do not evaluate the implications of having limited communication channels. In this paper, we propose Attentive Regularized Communication (ARCOMM), a new method for communication in MARL. The proposed method uses an attention module to evaluate the weight of the messages generated by the agents, together with a message regularizer that facilitates learning more meaningful messages, improving the performance of the team. We further analyse how ARCOMM reacts to situations where the messages must be compressed before being sent to other agents. Our results show that the proposed method helps, through the power of communication, to improve the performances of the agents in complex domains when compared to other methods. Furthermore, we show that, although there is a decrease of performance, agents are still capable of learning even with lossy communication. The messages learned by the agents also support the motivations for our method.

## 1 Introduction

In Multi-agent systems, agents often face situations where they can benefit from information that the others are observing somewhere else at a certain moment, but that they cannot directly see (Liu et al., 2021; Foerster et al., 2016). In such cases of partial observability, communication can be one of the key factors to enable agents to cooperate as a team (Kim et al., 2021). If communication is available, then they can share some of their perceptions of the surroundings with the rest of the teammates.

Along these lines, communication in Multi-Agent Reinforcement Learning (MARL) has become a popular subject of research lately (Das et al., 2019; Sukhbaatar et al., 2016; Liu et al., 2021; Foerster et al., 2016). In traditional communication in MARL, many of the proposed methods are built together with other standard MARL methods that operate under the centralised training with decentralised execution paradigm (CTDE) (Oliehoek et al., 2008; Kraemer & Banerjee, 2016). In this setting, agents have access to extra central information during training but are then restricted to their local observations when they execute their policies. Despite the promising results of operating under this configuration, this becomes unrealistic when we consider some practical applications, mainly because it is often impossible to let the agents access central information of the environment coming from a centralised oracle (Canese et al., 2021). Communication approaches can be seen as a way of alleviating some of these limitations by allowing the agents to communicate with each other (Du et al., 2021; Wang et al., 2023). When using the traditional CTDE paradigm, the agents do not benefit from communication only during the training stage. If communication is available, the execution phase can also be improved when agents can broadcast what they see or know at a certain moment to the rest of the teammates. It is important to note that this is still decentralised execution since they only share what they see and not something that is necessarily considered central information. In addition, communication does not share the raw values, but a certain encoding that is learned by the agents. By doing so some encoded messages are being broadcasted by the agents that containt their individual perceptions. However, it is important to ensure that the messages have enough quality to enable efficient cooperation.

Another important factor in communication-based methods is the bandwidth available for communication at a certain moment (Resnick et al., 2020; Tung et al., 2021). In the same way that in the context of wireless communications entities often need to compress their messages so that these fit into a restricted band (Mohamed, 2022), also agents might encounter scenarios where these constraints are present. In such cases, it is important that the agents are capable of compressing their messages.

In this paper, we consider the setting of communication in MARL under the CTDE paradigm. We intend to investigate how to enable agents to produce more meaningful messages in the communication process while also showing consistently strong performances across different scenarios. To this end, we propose **A**ttentive **R**egularized **Comm**unication (ARCOMM), a communication method for MARL that can be used together with any value function factorisation method. We show how this method improves learning in complex tasks by allowing the agents to exchange meaningful messages. Prompted by the recent success of attention-based communication architectures in MARL (Das et al., 2019; Jiang & Lu, 2018), ARCOMM starts by using an attention module to create a richer message encoding, to which we add a message regularizer that stabilizes the messages learned to contain more valuable information about the observations of the agents.

Considering cases where communication channels are limited, we also analyse how messages can be compressed from a different perspective. More specifically, we investigate how lossy information-based compression through the Discrete Cosine Transform (DCT) can be applied to the messages created by the agents to reduce their sizes, and whether this type of message compression is effective. Importantly, if this compression scheme proves to be effective, then it can be further used in other applications that require messages to fit into a constrained channel. Other related communication methods often fail to deal with compression and learn efficient communication strategies (Wang et al., 2019).

## 2 BACKGROUND

### 2.1 DECENTRALISED PARTIALLY OBSERVABLE MARKOV DECISION PROCESSES (DEC-POMDPS)

In this work, we model the learning problems following Decentralised Partially Observable Markov Decision Processes (Dec-POMDPs) (Oliehoek & Amato, 2016). The Dec-POMDP can be defined as a tuple $G = \langle S, A, O, Z, P, r, \gamma, N \rangle$, where $s \in S$ represents the current state of the environment. From the current state, local observations $o_i$ can be derived according to a function $O(s, i) : S \times \mathcal{N} \rightarrow Z$ for a certain agent $i \in \mathcal{N} \equiv \{1, \ldots, N\}$. Additionally, the agents also maintain an action-observation history $\tau_i \in \mathcal{T} \equiv (Z \times A)^* \rightarrow \{\tau_1, \ldots, \tau_N\}$. When in a given state, each agent performs an action $a_i \in A$ where $A$ is the action space, forming a joint action $a = \{a_1, \ldots, a_N\}$ that is executed in the current state $s$ of the environment, and from which it results a reward that is shared by the entire team, $r(s, a) : S \times A \rightarrow \mathbb{R}$. After the actions are executed, the environment transits to a next state $s'$ according to a probability function that models the dynamics of the environment, $P(s'|a, s) : S \times A \times S \rightarrow [0, 1]$. Usually, the actions of the agents are controlled by a policy $\pi_i(a_i|\tau_i) : \mathcal{T} \times A \rightarrow [0, 1]$, but in the considered context of communication in MARL, the policy is instead conditioned not only on $\tau_i$, but also on a set of incoming messages from the other agents $m_{-i}$, meaning that the corresponding policy that controls the actions of the agents can be written as $\pi_i(a_i|\tau_i, m_{-i})$ (where $-i$ refers to all except $i$). During learning, the joint objective of the agents is to maximise an action-value function $Q_\pi(s_t, a_t) = \mathbb{E}_\pi[R_t|s_t, a_t]$, where $R^t = \sum_{k=0}^{\infty} \gamma_k r_{t+k}$ is the discounted return with a discount factor $\gamma \in [0, 1)$.

### 2.2 CENTRALISED TRAINING WITH DECENTRALISED EXECUTION (CTDE) AND COMMUNICATION IN MARL

Within MARL, centralised training with decentralised execution (CTDE) is a popular paradigm that allows the agents to be trained in a centralised manner, but they must execute their policies in a decentralised way. Communication-based methods can be easily integrated into this learning structure. The key difference from simple CTDE is that, when we use communication, the agents can broadcast information to others not only during training but also during execution. This means that, despite the learned policies being decentralised, the agents can receive encoded messages that represent what their teammates see or sense at a certain timestep. While in the fully centralised setting there is a

common oracle that can see everything in the environment, a communication setting can be seen as a more realistic choice since communication is done on a peer-to-peer basis where each agent is responsible for broadcasting its own messages. This way, they are not dependent on a central unit that has the big - and often unrealistic - advantage of observing everything at the same time.

Intuitively, communication in MARL should improve the performances of the agents when it is combined with other base methods, as shown in works such as Liu et al., 2021. If the messages learned are informative enough, they should be useful for the agents to learn to "talk" with each other and improve their cooperative strategies to solve complex tasks. However, if the messages learned are not adequate, this might result in an overload for the learning networks and harm the performances of the agents.

## 2.3 VALUE FUNCTION FACTORISATION IN MARL

In cooperative MARL, value function factorisation methods form a group of powerful algorithms to solve complex MARL tasks. The key idea of these methods is to learn a way of decomposing a joint action-value function into agent-wise functions (Sunehag et al., 2017),

$$Q_{tot}\left(\tau, a\right) = f\left(Q_i(\tau_i, a_i; \theta_i)\right), \forall i \in \{1, \dots, N\}, \tag{1}$$

where $f$ here represents a certain function that mixes the individual functions into a joint function $Q_{tot}$. An efficient decomposition of the joint action-value function should be done in a way that satisfies the Individual-Global-Max (IGM) condition (Son et al., 2019). This condition states that the set of local optimal actions should also maximise the joint Q-function. This can be formalised as

$$\arg\max_{a} Q_{tot}\left(\tau, a\right) = (\arg\max_{a_1} Q_1(\tau_1, a_1), \dots, \arg\max_{a_N} Q_N(\tau_N, a_N)). \tag{2}$$

The first value function factorisation method was proposed by Sunehag et al. (2017), which introduces Value Decomposition Networks (VDN) as a way of factorising the joint $Q_{tot}$ as the sum of the individual Q-functions. Later on, QMIX (Rashid et al., 2018) proposes a new non-linear way of factorising the $Q_{tot}$ that extends the range of functions that can be represented by VDN to a larger family of monotonic functions. Both these factorisation methods are sufficient to satisfy (2).

In this type of methods, the loss used to update the networks is based on the temporal difference loss as described in DQN (Mnih et al., 2015) that uses a replay buffer and a target network, but here with respect to a $Q_{tot}$. This loss can be formalised as

$$\mathcal{L}(\theta) = \mathbb{E}_{b \sim B} \left[ \left( r + \gamma \max_{a'} Q_{tot}(\tau', a'; \theta^-) - Q_{tot}(\tau, a; \theta) \right)^2 \right], \tag{3}$$

for some sample $b$ that is sampled from the replay buffer $B$, and where $\theta$ and $\theta^-$ are the parameters of the learning network and of a target network, respectively.

In this paper, we use VDN and QMIX to demonstrate our communication approach on top of standard MARL approaches that don't initially use communication.

## 2.4 DISCRETE COSINE TRANSFORM (DCT)

In the fields of data compression and signal processing, the Discrete Cosine Transform (DCT) (Ahmed et al., 1974) is a certain function based on a sum of cosine functions that processes a sequence of data points and encodes them into a compressed representation. By doing so, it is possible to achieve a much smaller representation of the data in terms of size occupied. For simplicity, we introduce here only the expression for the type II of the DCT (that is the most common form and the one used in this paper) that, generally, can be done following (Makhoul, 1980)

$$X(k) = 2 \sum_{n=0}^{C-1} x(n)\cos\left(\frac{\pi(2n+1)k}{2C}\right), 0 \leq k \leq C-1, \tag{4}$$

where $X(k)$ represents the $k^{th}$ transform of the DCT, and $x(n)$ denotes the sequence of values to be compressed, with size $C$. In the context of this paper, the DCT is applied to the last dimension of the vectors containing the messages of the agents, forming a compressed representation of their messages. Using this smaller representation is important when we consider channel sizes or

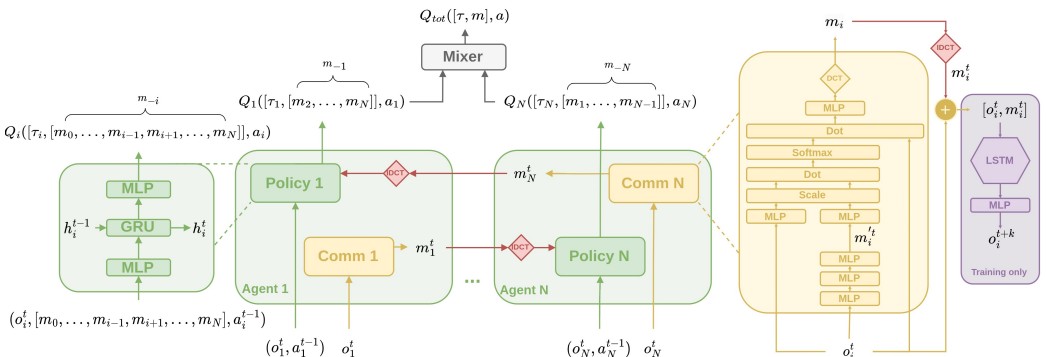

Figure 1: Architecture of the proposed communication method for MARL. The proposed method can be used together with any value function factorisation method (whose mixer is represented in the figure in the block *mixer*) and uses parameter sharing. The pink diamonds represent the inverse DCT that is applied to decompress the messages that are compressed with the DCT (yellow diamond) after they are computed by the communication network (yellow). Note that both the DCT and IDCT blocks are only used when we analyse the effect of compression in section 4.2. In the main experiments, these blocks are not applied.

other space constraints, through which only compressed representations can be sent. When this representation arrives at the other end of the communication channel, the inverse of the function is applied, and the message can be recovered with some level of information loss. We hypothesise that, if this information loss is not too heavy, the agents can still learn and benefit from communication in MARL. This decompressing process can be done by inverting Eq. (4). The inverse can be calculated following (Makhoul, 1980)

$$x(n) = \frac{1}{C} \left[ \frac{X(0)}{2} + \sum_{k=1}^{C-1} X(k) cos \left( \frac{\pi(2n+1)k}{2C} \right) \right], 0 \le n \le C - 1. \tag{5}$$

The DCT can be used both as a lossy and as a lossless compression method, depending on the parts of the messages that are encoded. In this paper, we consider the case of lossy compression, since for lossless compression to be achieved the messages are encoded but without reducing their sizes, which becomes redundant when we consider limited communication channels.

## 3 METHODS

### 3.1 COMMUNICATION IN MARL: ATTENTIVE REGULARIZED COMMUNICATION (ARCOMM)

In this section, we propose ARCOMM, a new architecture for efficient communication in MARL. The idea is to build an inter-agent communication architecture that is capable of learning meaningful messages to ensure cooperation in complex MARL tasks. ARCOMM starts by using an attention module to learn messages that are weighted among agents and contain richer information. In the proposed architecture, the messages are generated from the local observations of the agents. We start by applying a set of linear operations to the initial observations as a first encoding. Secondly, these encodings are given as an initial message $m'$ to an attention module, in order to learn their relative importance. These values are embedded into $k \in \mathbb{R}^{d_k}, v \in \mathbb{R}^{d_k}, q \in \mathbb{R}^{d_k}$, where $d_k$ is the embedding dimension. Importantly, the messages used in the attention module come from an initial embedding that is applied in the observations (as shown in Fig.1). In this sense, we define the keys, queries, and values, at each timestep $t$, for the attention operations as

$$k_t = \left[ W_{K,1} \tau_1^t, \dots, W_{K,i} \tau_i^t, \dots, W_{K,N} \tau_N^t \right], \tag{6}$$

$$v_t = \left[ W_{V,1} \tau_1^t, \dots, W_{V,i} \tau_i^t, \dots, W_{V,N} \tau_N^t \right], \tag{7}$$

$$q_t = \left[ W_{Q,1} m_1'^t, \ldots, W_{Q,i} m_i'^t, \ldots, W_{Q,N} m_N'^t \right], \tag{8}$$

where $W_{Q,i}, W_{K,i}, W_{V,i}$ are trainable weight matrices, and where $m'^t$ corresponds to the initial message embeddings that will be refined (as in Fig. 1). As a result, the attention weights calculated for an agent $i$ from the messages and observations in our approach can be formalised as

$$\alpha_{ij} = \frac{\exp(\phi \cdot (q_i^t \cdot k_j^{t\,T}))}{\sum_{x \in \mathcal{N}} \exp(\phi \cdot (q_i^t \cdot k_x^{t\,T}))}, \tag{9}$$

where $\phi$ is a scaling factor. These weights are further used to calculate a new aggregated attention-encoded message $m_i^t$,

$$m_i^t = \sum_{j=1}^{N} \alpha_{ij} v_j^t, \tag{10}$$

where the weights $\alpha_{ij}$ define the relationship between agents $i$ and $j$, and hence the importance of value $v_j^t$. The intuition behind the choice of keys, queries and values, is that, by relating the messages to multiple different latent representations of the observations, the messages learned will be able to capture more relevant information from the observations.

Despite the recent success of attention-based architectures for multi-agent communication in complex scenarios, we hypothesise that the messages originated by our base architecture are not rich enough to learn complex environments. In this sense, we build an additional layer of complexity to our communication architecture by proposing a message regularizer. The key intuition is that, if the messages generated by the agents help to predict their own next observations, it means that these messages are likely to contain more meaningful information about their individual observations. We hypothesise that the uncertainty associated with the future values of the observations is reduced when we use both the previous values of the messages and the observations, when compared to using only the previous values of the observations. As such, we use a recurrent encoder that receives the previous messages alongside the previous observations, $[o_i^{-k}, m_i^{-k}]$ with values up to $T - k$, and predicts the next $k$ observations ahead $o_i^{+k}$, where $k$ denotes the number of timesteps to predict ahead of each timestep $t$, and $T$ the total length of the episode. For our experiments, we predict only one timestep ahead of each $t$. Let $g(\cdot; \theta_r)$ here denote a certain neural network with parameters $\theta_r$ composed of an LSTM module that estimates the next values of the observations of the agents given the previous messages and the previous observations (as described in Fig. 1). The observations predicted by this network for the timesteps ahead can be given by $o_i^{+k'} = g([o_i^{-k}, m_i^{-k}]; \theta_r)$. The predicted outputs of this network can then be used to calculate a second loss that will auxiliate our learning problem, as described in

$$\mathcal{L}_m = \frac{1}{T - k} \sum_{t=1}^{T-k} \| o^{+k} - o_i^{+k'} \|_2^2, \tag{11}$$

for an agent $i$, where $T - k$ denotes the number of predicted values in the vector, given that an episode lasts $T$ timesteps and $g(\cdot; \theta_r)$ is predicting $k$ timesteps ahead of each timestep $t$. For the proposed method, this additional loss is used alongside the loss described in Eq. (3), resulting in the overall objective of minimising, with respect to $\tau$, the following loss

$$\mathcal{L}(\theta, \theta_c, \theta_r) = \sum_{b=1}^{B} \left[ \left( y_{tot} - Q_{tot}(\tau, a; \theta) \right)^2 + \frac{1}{T - t} \sum_{t=1}^{T-k} \| \tau^{+k} - g(\tau^{+k}, m^{+k}; \theta_r) \|_2^2 \right], \tag{12}$$

for batch of samples $B$, and where $y_{tot} = r + \gamma \max_{a'} Q_{tot}(\tau', a'; \theta^-)$, for the parameters of a network and the respective target, $\theta$ and $\theta^-$, the parameters of the communication network $\theta_c$, and the parameters of the regularizer module $\theta_r$. In the sections ahead we demonstrate how this message regularizer positively affects the learning process of the agents. Fig. 1 depicts the architecture of the proposed method as described in this section (note that the DCT and IDCT blocks are represented for completeness purposes and should be disregarded when compression is not being analysed). Importantly, we build our communication method on top of value function factorisation methods and adopt the famous parameter sharing convention (Gupta et al., 2017). In this sense, our method can be easily integrated with any existing value function factorisation method. In the experiments section, we show results for a set of these methods.

## 3.2 Message Compression for Lossy Communication in MARL

In this work, we also intend to study whether communication can still be done efficiently when the messages exchanged are, for example, lost, or dropped during the process. For that, instead of simply zeroing or naively cutting values of the messages, we use the DCT, a popular lossy compression method. Importantly, we note that this method for message compression does not require changing the sizes of any of the agent networks, since the compression only applies to a potential communication channel, and then the messages are reconstructed to the original size when they reach the other communication end.

In our architecture, each agent will generate its messages according to the method described in the previous subsection, and then, in the results presented in subsection 4.2, the generated messages will be compressed using the DCT, as described in Eq. (4). In this work, we focus on lossy compression, i.e., each agent compresses the message in a way that will make it smaller in size, reducing the potential communication overhead. When the messages arrive at the destination, these are decompressed using the inverse of the DCT (IDCT, as described in Eq. (5)). The integration of this process together with communication in MARL is depicted in Fig. 1. In the figure, we can see that the messages are generated using the network described in the previous subsection. When we consider message compression, the change in the architecture comes from the compression of the message after it is generated (DCT in the yellow diamond of the figure), and then it is decompressed in the destination (IDCT, illustrated by the pink diamonds in the figure). Finally, in section 4.2 we use the result of this entire process as input of the agents, together with their own observations, in the same way that was described before, but now the messages are compressed in the source and then decompressed in the endpoint. This results in the input $[\tau_i, m^*_{-i}]$ for agent $i$, where $m^*_{-i}$ here represents the decompressed messages coming from the other agents.

## 4 Experiments and Results[1]

In this section, we present the experiments carried out to evaluate the performance of the proposed ARCOMM. While the proposed communication method can be built on top of any value function factorisation method in MARL, in this paper we apply it together with the architectures of QMIX (Rashid et al., 2018) and VDN (Sunehag et al., 2017), two popular value function factorisation methods in MARL, as introduced in subsection 2.3. We demonstrate the performances of ARCOMM in comparison with the vanilla methods without communication, and two popular communication methods, COMMNET Sukhbaatar et al., 2016 and TARMAC Das et al., 2019. Note that, in the results in 4.1, there is no compression, i.e., the DCT and IDCT blocks in Fig. 1 are not used. These are only used when investigating message compression in 4.2. Hyperparameters and other details can be found in Appendix E.

### 4.1 Main Results with ARCOMM

Fig. 2(a) illustrates the performances in one of the SMAC scenarios (Samvelyan et al., 2019), where the team of agents is composed of 3 stalker units (ranged) that must defeat a team composed of 5 zealots (melee). Considering that this is the simplest scenario out of all the experimented ones, it was expected that all the methods would be able to solve it somewhat easily, as the figure shows. Particularly in the case of QMIX, we can see that the agents benefit from communication, making them solve the task sooner. However, in the case of VDN, it doesn't show to be as useful. This suggests that, for this scenario, communication might not be as important as in other cases to learn efficient strategies, although convergence is still achieved quickly.

Yet, we can see that, in the other more complex scenarios, communication proves to have a stronger impact. In Fig. 2(b), we can see that, for 2c_vs_64zg, ARCOMM enables the agents to learn the task much faster and at a higher level of performance. This map models a scenario where 2 colossus units must defeat a team of 64 zerglings Samvelyan et al., 2019. Both QMIX and VDN show much-improved performances when combined with ARCOMM, with particular emphasis on VDN, where the improvement is outstanding. Finally, in the case of MMM2, in Fig. 2(c) we can also see benefits of communication, although these are not as prominent as in the case of the previous scenario.

---

[1]Codes available at https://github.com/mrpnbtr/arcomm-marl

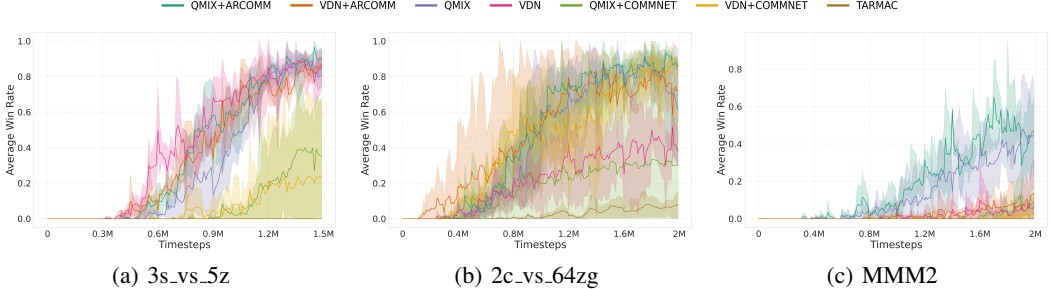

Figure 2: Performance of the attempted methods in the Starcraft environments. The plots depict the average win rates of the agents over training time. (a) In 3s_vs_5z 3 stalker units must defeat a team with 5 zealot units. (b) In 2c_vs_64zg, the agents are 2 colossi playing against 64 zerglings. (c) MMM2 is a complex map where 1 medivac (healing unit), 2 marauders and 7 marines play against 1 medivac, 3 marauders and 8 marines.

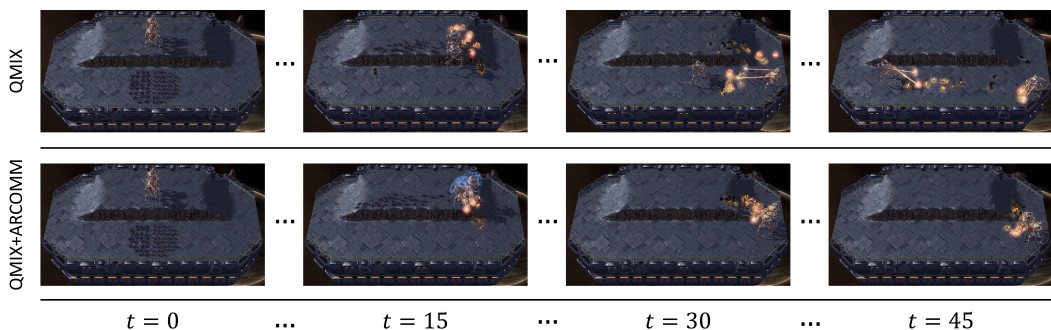

Figure 3: Learned behaviours for QMIX (top) and QMIX+ARCOMM (bottom) after training in 2c_vs_64zg. When we use ARCOMM, we can see that the agents will be always cooperating close to each other, while without communication they eventually move away from one another (top right).

The effect of communication becomes evident mostly when we look at QMIX, where ARCOMM has a strong positive impact, allowing to learn the task sooner. Even in the case of VDN, we can still see improvements, although the performance stays below. When we look at other communication methods, interestingly, we can see that, in general, TARMAC (Das et al., 2019) does not perform well in these complex tasks, while COMMNET (Sukhbaatar et al., 2016) shows a reasonable performance. However, we note the inconsistent performance of COMMNET, which performs reasonably in some cases (Fig. 2(a) and 2(b)), but then completely fails in others (Fig. 2(c)). Taking a deeper look at the behaviours of the agents, the positive impact of using communication with ARCOMM is further illustrated in Fig. 3 for 2c_vs_64z. We can see in this figure that agents that use ARCOMM to communicate (bottom) tend to remain cooperating close to each other for the entire episode, while when they do not communicate (top), they eventually start moving away from each other.

To further demonstrate the strength of the proposed method, we evaluate ARCOMM in a PredatorPrey environment (Koul, 2019). Previous works such as (Son et al., 2019; Liu et al., 2021; Böhmer et al., 2019) have demonstrated the importance of considering these scenarios that impose stronger punishments for non-cooperative behaviours. In our experiments, we consider a version of this environment where 4 agents must catch 2 moving prey in a $7 \times 7$ grid. At least two of them are needed to catch one prey, and when they do, the team receives a reward of $5 \times N$, where $N$ is the number of agents. Each step there is a small penalty of $-0.1 \times N$ and, most importantly, there is a team penalty $p \times N$ that punishes the agents when one of them attempts to catch a prey alone. In Fig. 4 we can see the rewards achieved by the experimented methods for different values of $p$. When we use ARCOMM, the agents take advantage of the messages sent by the others and can solve the task by understanding what the others experience. While for $p = -0.5$ communication is not yet very important, as we increase to $p = -0.75$ (Fig. 4(b)) and $p = -1.0$ (Fig. 4(c)), we can see that

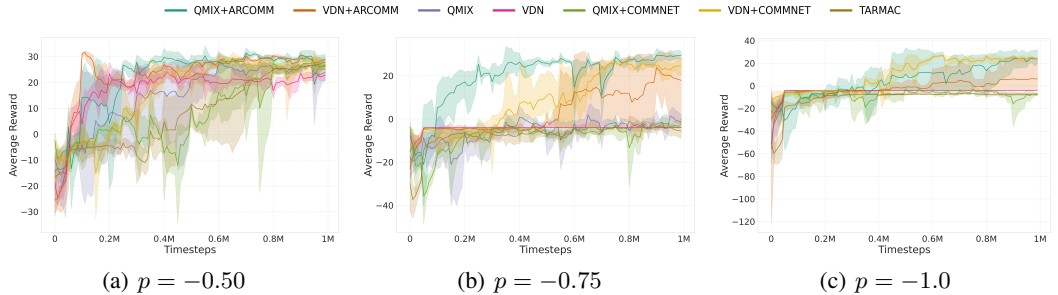

(a) $p = -0.50$        (b) $p = -0.75$        (c) $p = -1.0$

Figure 4: Performance of the attempted methods in the PredatorPrey environment for different levels of cooperation penalty. The plots depict the average rewards of the agents over training time.

communication is necessary, and the agents cannot solve the task without it. Interestingly, while TARMAC only proves to be helpful for $p = -0.75$, when we scale to $p = -1.0$ COMMNET takes the lead, although ARCOMM reaches the same level later. Importantly, we observe the inconsistency of methods like COMMNET and TARMAC, which do well in some cases, but completely fail in others. On the other hand, ARCOMM shows consistently good performances in all the experimented scenarios. Further results can be found in Appendix B.

## 4.2 ADDITIONAL RESULTS: COMPRESSING MESSAGES

In the previous section, the results presented assume optimal conditions where communication among entities can be done in a lossless manner, i.e., where messages can be exchanged without loss of information. In this subsection, we present the results of communicating under a lossy communication channel, using the same proposed architecture. More specifically, we use the DCT to compress the messages into a smaller representation. Importantly, this does not affect the size of the communication endpoints, since the messages are reconstructed once they reach the destination, and thus there is no need to modify the size of the networks for different levels of compression.

Fig. 5 illustrates the performances of VDN+ARCOMM under different levels of DCT lossy message compression. In the figure, we can see that VDN+ARCOMM can still learn even when using compressed messages, despite there is also a decrease of performance in both of the attempted scenarios. However, in 5(b), there is still a big increase of performance when compared to VDN without communication in this environment (Fig. 2). Importantly, this means that, even when using compressed and hence lightweight messages, the performance can still be improved with communication when it is necessary. In Appendix D we provide additional experiments to analyse the effect of message compression.

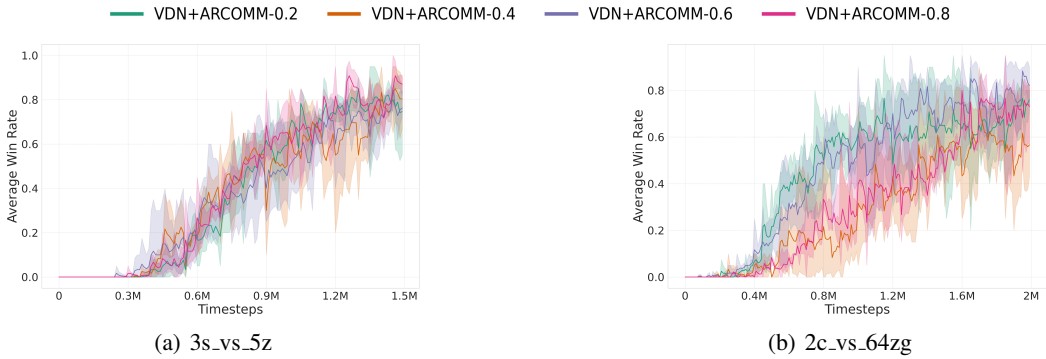

(a) 3s_vs_5z            (b) 2c_vs_64zg

Figure 5: Results for the communication approaches when we use the DCT to compress the messages that the agents generate. To evaluate the impact of the lossy compression of the DCT, we compress the messages to 20%, 40%, 60%, and 80% of their original size, as illustrated in the figure.

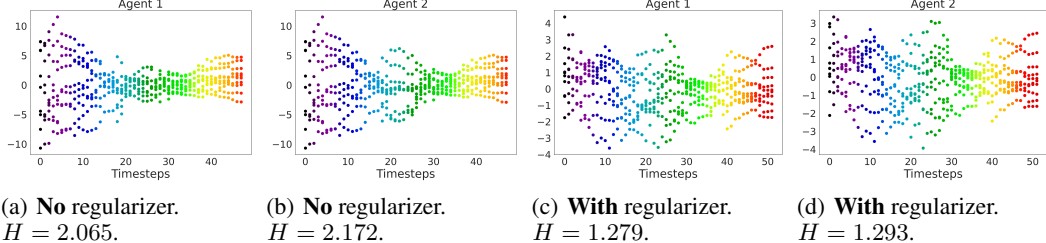

(a) **No** regularizer. $H = 2.065$.

(b) **No** regularizer. $H = 2.172$.

(c) **With** regularizer. $H = 1.279$.

(d) **With** regularizer. $H = 1.293$.

Figure 6: Messages learned by the two agents in 2c_vs_64zg throughout one successful episode with QMIX+ARCOMM when the message regularizer is removed (a-b), and when it is used (c-d). The values of $H$ denote the entropy of the messages.

### 4.3 ABLATIONS: MESSAGE REGULARIZER

We have previously described how ARCOMM uses a message regularizer to improve the quality of the messages learned during training. In this subsection, we analyse the impact of the message regularizer on the messages produced in the learning problem.

To analyse the messages learned, we have saved the trained networks of QMIX+ARCOMM for the 2c_vs_64zg task, both using and not using the regularizer (purple box in Fig.1). Fig. 6 shows the messages produced by the two agents involved in this task during a successful episode, i.e., an episode where the agents win the game. We can see in Fig. 6(c) and 6(d) that the messages produced when the agents use the message regularizer are compact within a smaller interval (around $[-4, 4]$) when compared to the messages produced when we do not use the regularizer (Fig. 6(a) and 6(b)). In the latter case, the messages produced assume values that lie inside a much larger range (around $[-10, 10]$), meaning that the possible kind of messages learned by the agents is more ambiguous, making the task more difficult for them since the messages will not be as precise and meaningful as in the case of the message regularizer. The values of the entropy $H$ for the messages produced that are shown in the caption of the figures also support our observations that messages that were produced with the help of the regularizer will contain less uncertainty and ambiguity.

## 5 CONCLUSION AND FUTURE WORK

Communication approaches in MARL have shown to provide an efficient way of learning cooperative strategies between agents. When agents understand what the others are experiencing, they can learn how to accommodate their behaviours and anticipate situations in such a way that the whole team will benefit. In this paper, we have proposed ARCOMM, a new method for communication in MARL. ARCOMM can be used together with any value function factorisation method, and the experiments carried out in this paper demonstrate how this method facilitates solving complex tasks in MARL. While in some scenarios the benefits of communication might not be as evident, in others it is clear how it drastically improves learning. At the same time, by analysing the type of messages learned by ARCOMM, we observed that the proposed message regularizer can be important in improving the type of messages learned, making them more meaningful to the agents.

In multiple practical situations, communication channels with limited bandwidth can be faced and hence there is a need to study whether we can still leverage the power of communication under such conditions. In this work, we have investigated how compressing the messages produced by the agents with the DCT affects their performances in complex tasks. The results have shown that communication can still be beneficial, even when messages need to be compressed.

In the future, we aim to explore the impact of communication when we use different capacities for the agent and communication networks. In addition, we intend to study how compressing the messages affects learning also under different network sizes, and how it can be extended to practical scenarios.

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

# A    RELATED WORK

In this section, we briefly present further relevant work related to this paper. MARL has been well studied in the recent past (Liu et al., 2021; Wang et al., 2021; Sunehag et al., 2017). Particularly in cooperative MARL, value function factorisation methods represent a branch of popular algorithms whose core objective is to learn a decomposition of a joint Q-function into agent-wise Q-functions. Numerous approaches that follow this configuration have shown outstanding results in multiple complex scenarios (Rashid et al., 2018; SHEN et al., 2022). Importantly, These methods operate under the well-known CTDE paradigm. While this configuration represents the middle of the learning spectrum, the other two ends have also been studied, but it has been discussed that neither full centralisation nor full decentralisation can achieve as good results (Sunehag et al., 2017).

As an alternative to the conventional methods, communication in MARL has gained the attention of the scientific community (Liu et al., 2021; Jiang & Lu, 2018; Foerster et al., 2016; Das et al.; Zhang et al., 2019; 2020). At its core, the key difference in these methods when compared to conventional approaches is that agents can share something about their experience of the environment, both during training and execution. In (Foerster et al., 2016) the authors have introduced one of the very first communication-based MARL methods. In simple terms, it is shown how two agents can solve tasks where they must communicate some of what one knows but the other doesn't. In (Sukhbaatar et al., 2016), the authors show the effect of communication in more complex scenarios with more agents. In particular, it is shown that, when agents communicate, their level of efficient cooperation increases in scenarios with more complex objectives, such as traffic junction negotiations. As these approaches became more popular, more complex methods started to arise. These consider factors such as what, whom or when to communicate. For example, in (Jiang & Lu, 2018) it is proposed an attention-based mechanism that allows the agent to decide to whom they should send their messages. In the case of (Das et al., 2019), the authors propose an improved messaging mechanism that consists of signing the messages, allowing to target specific agents with customised information. Also in (Chu et al., 2020), the authors propose a communication method that uses message fingerprints and a different way of aggregating all the messages of the agents. What is shared can also be important, as demonstrated in works such as (Liu et al., 2021) or (Kim et al., 2021), where agents that share their intentions achieve better performances.

Still involving MARL but on a slightly different scope, language discovery in communication games has also been aim of studies. Usually in games that involve a speaker and a listener agent, the kind of language that emerges during the communication process has been studied in works like (Mordatch & Abbeel, 2018), where the agents learn a task-specific vocabulary during training that they use to solve the tested games. Also in (Gupta et al., 2020) or (Kajic et al., 2020) the authors evaluate the interpretability of the languages learned by the agents in terms of how humans can interpret them. Another step towards interpretability is to ground language, as it is studied in (Lin et al., 2021) where the authors use an autoencoder module that ground language by reconstructing observations. While this deeper analysis becomes more challenging as more agents are used, it is still important to analyse what the agents devise to communicate in order to understand their learning process and how they understand the imposed tasks.

# B    ADDITIONAL RESULTS: OTHER SCENARIOS

To support the results presented in section 4.1, we demonstrate in this section the strength of the proposed in other relevant environments. We test our method in Lumberjacks (Fig. 7(a)), TrafficJunction (Fig. 7(b)) and 3s5z (SMAC). As discussed in the main paper, in some SMAC environments communication might not be as important as in other scenarios. Thus, in addition to the PredatorPrey results in 4, we carried additional experiments in other complex environments. However, we still stress the importance of these methods being able to solve a wide range of environments and not only scenarios where communication is needed.

**Lumberjacks** (Fig. 7(a)) consists of an environment where 4 agents must chop all the existing 12 trees in a $8 \times 8$ map. In this environment, the agents receive a penalty of $-1$ every timestep and a reward of $+10$ when a tree is cut Koul (2019). Each tree is assigned a random level between 1 and the number of agents, where this level represents the number of agents required at the same time to

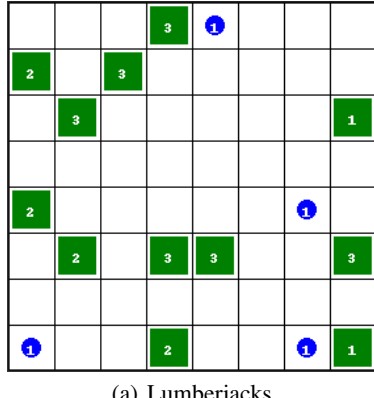

(a) Lumberjacks

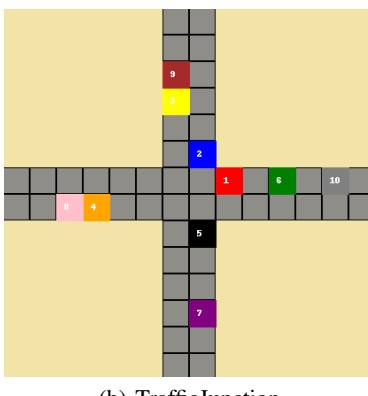

(b) TrafficJunction

Figure 7: Visual representation of 2 of the additional environments experimented with (from Koul (2019)).

cut the tree. Importantly, the high step penalty and the elevated need for cooperation make this task challenging.

**TrafficJunction** (Fig. 7(b)) represents a cross-shaped traffic junction where agents coming from 4 different entries in the junction must cross to a pre-defined location at the end of another road after crossing the junction Koul (2019). In our case, we have defined the number of agents to 10 agents, making this task very challenging. The agents received a penalty of $-0.01$ to incentivate them to keep moving and a penalty of $-10$ if they collide with another agent.

**3s5z** is another environment from the SMAC collection that we use to further demonstrate that the proposed method is robust across different types of tasks, both in cases where communication is necessary and where it might not be utterly necessary. Additionally, the team in this environment is composed by 8 agents of 2 differnt types, making it a challenging map within the collection.

Fig. 8 shows that the proposed method has improved performance across all scenarios. In 3s5z (Fig. 8(c)) we observe once again that the improvements are not very significant since communication might not be as important in this case. However, there is still an improvement of performance, mainly at the beginning of training where VDN+ARCOMM lerns the task before the other methods. In the other environments in Fig. 8(a) and Fig. 8(b) we observe a different scenario. In these complex environments communication shows to have a strong impact and the proposed method demonstrates substancially improved performances over the baselines. This kind of environments has been used in other works to evaluate the strength of communication, such as in (Sukhbaatar et al., 2016). Importantly, we note once again the robustness of ARCOMM across environments, while others might work well in some of them, but perform poorly in others.

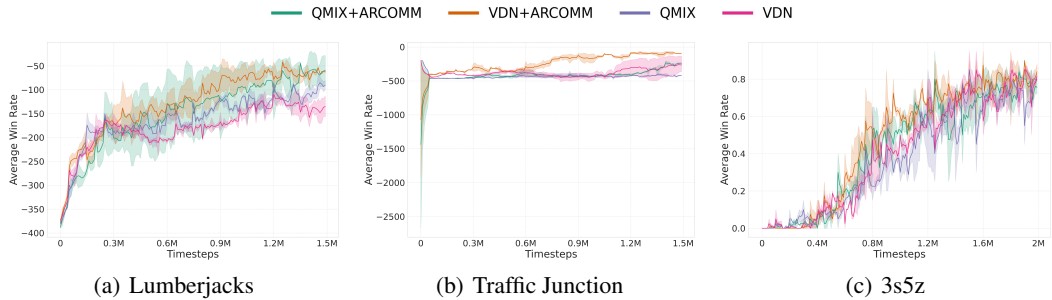

(a) Lumberjacks

(b) Traffic Junction

(c) 3s5z

Figure 8: Performance in (a) Lumberjacks, (b) Traffic Junction and (c) 3s5z (SMAC). The plots (a) and (b) depict the average team reward over time and in (c) it is depicted the average win rates.

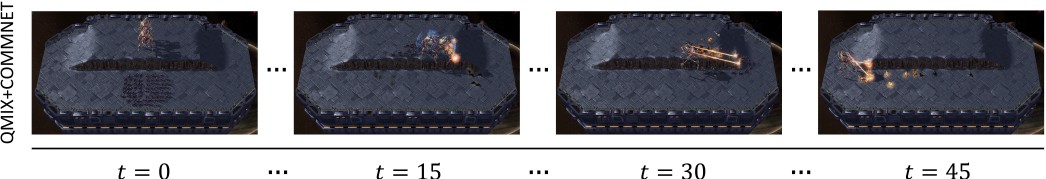

Figure 9: Learned behaviours for QMIX+COMMNET after training in 2c_vs_64zg. One of the agents is destroyed after the team spends a long time in the middle of the enemies.

## C  ADDITIONAL RESULTS: LEARNED BEHAVIOURS

In section 4.1 we have analysed how the agents behave when they take advantage from communication when compared to not communicating. More specifically, we have compared the behaviours of QMIX+ARCOMM with simple QMIX. For completeness, we show in this section the behaviours for another method that uses communicaiton, QMIX+COMMNET, inn the same environment, 2c_vs_64zg. In the figure below, our observations are in line with the previous analysis, where agents tend to be together more often when they communicate, as with ARCOMM. However, in this case the agents assume a less optimal strategy where they spent long periods of time together in the middle of the enemy team, losing health points and resulting in the death of one of the agents, leaving the other alone to finish the game. This suboptimal behaviour does not happen when we use ARCOMM and this is reflected in the team performances in Figure 2.

## D  ADDITIONAL RESULTS: MESSAGE COMPRESSION

In addition to the effects of compression on VDN+ARCOMM that we have shown in section 4.2 in the main paper, we extend here the experiments to evaluate the effect of compression also on QMIX+ARCOMM. Fig. 10(a) depicts the performances of QMIX+ARCOMM under different levels of DCT lossy message compression. As expected, we can see that the performances will decrease as the level of message compression increases. However, it is important to note that the agents can still learn the tasks despite the high levels of compression. In some cases where communication channels might be constrained to a certain bandwidth, it is important to ensure that communication can still be leveraged to improve cooperation in MARL.

To investigate how the compression method affects the messages learned by the agents, we have looked at the messages devised by the agents for a 40% level of compression with QMIX+ARCOMM in the 2c_vs_64zg environment. In Figure 11 we can see that the messages will occupy smaller ranges, than when compared to when there is no compression (illustrated previously in Fig. 6). This suggests that the agents try to find more specific messages with less broad values when these must be compressed afterwards.

## E  IMPLEMENTATION DETAILS AND HYPERPARAMETERS USED IN THE EXPERIMENTS

In the experiments carried out in this paper, the agents are controlled by a deep recurrent Q-network that uses a GRU (gated recurrent unit) with width 64. Additionally, in QMIX the hidden layers used by the mixer have size 32. All the experimented methods share the learning networks, following the parameter sharing convention widely adopted in the literature. This allows to speed up learning but requires that an agent ID is added to the inputs of the agents so that the network can differentiate them. Regarding the communication networks in ARCOMM, we set the size of all the embeddings of the attention mechanism to 64. The length of the messages that each agent produces at each timestep is defined as 10. When using compression, a value of, for example, 40% of compression will naturally result in messages of size 6, and so on. Regarding TARMAC, we use it with QMIX.

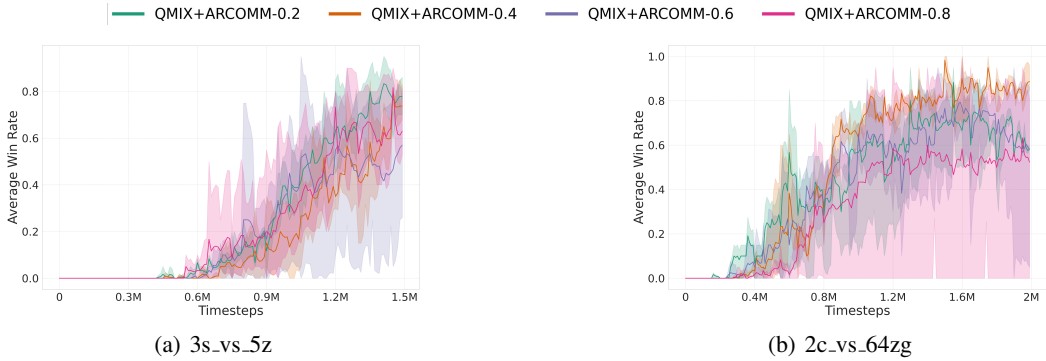

(a) 3s_vs_5z
(b) 2c_vs_64zg

Figure 10: Results for the communication approaches when we use the DCT to compress the messages that the agents generate. To evaluate the impact of the lossy compression of the DCT, we compress the messages to 20%, 40%, 60%, and 80% of their original size, as illustrated in the figure.

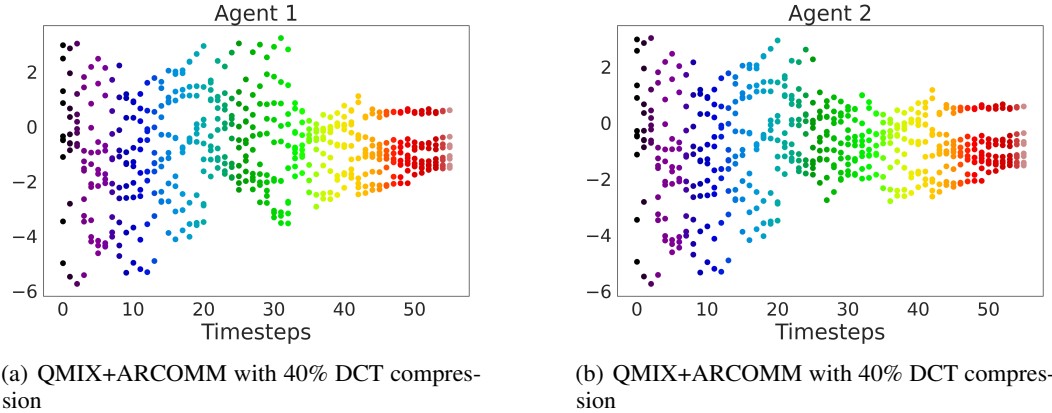

(a) QMIX+ARCOMM with 40% DCT compression
(b) QMIX+ARCOMM with 40% DCT compression

Figure 11: Messages generated by the two agents over the course of a successful 2c_vs_64zg episode, after being trained with QMIX+ARCOMM under a message compression level of 40% with the DCT (messages plotted after being decompressed at the destination using the IDCT).

We use a replay buffer to store experiences with size 5000, from which minibatches with size of 32 episodes are sampled for training. The replay buffer is updated over time. To take actions, the exploration-exploitation trade-off of the agents follows the epsilon-greedy method, with the value epsilon, $\epsilon$, starting at 1. This value anneals gradually throughout 50000 training episodes down to a minimum of 0.05. The value of the discount factor $\gamma$ is set to 0.99. The parameters of the target networks are updated every 200 episodes. All the networks are trained using the RMSProp optimization algorithm, with a learning rate $\alpha = 5 \times 10^{-4}$.

