# OpenReview forum: "Learning Multi-Agent Communication using Regularized Attention Messages"
_ICLR.cc/2024/Conference — Submitted to ICLR 2024_

### Official Review · Reviewer_eV4L · 2023-10-30

**Soundness:** 2 fair
**Presentation:** 2 fair
**Contribution:** 2 fair
**Rating:** 5
**Confidence:** 4

**Summary:**

Despite numerous successful demonstrations of the benefits of communication in solving cooperative tasks with multi-agent reinforcement learning (MARL), these methods could still struggle in complex scenarios like when communication channels have limited capacity. The paper proposes an attention module to weigh messages with a message regularizer to induce more meaningful messages. Experimentally, the proposed method is shown to improve performance and is more robust against situations when there is lossy communication.

**Strengths:**

1. The proposed method shows some improvement over the baselines used
2. The lossy communication is an important one to investigate.
3. The method section is clear.

**Weaknesses:**

1. Imprecise lanugage, writing needs improvement:
    * Second paragraph of introduction is a bit repetitive
    * Meaning of ‘quality’ in message is vague. What are messages with sufficient quality to enable efficient cooperation?
    * ‘an overload for the learning networks’: what does this mean?
2. The use of an attention module for communication is not that novel. In addition to the cited work, [1] is also a similar work.
3. The message regularizer resembles AEComm [2], reconstructing observation, which is not cited
4. Results have very high variance, making the conclusions weak. The improvement does not seem to apply across methods. I disagree with the statement that ARComm is better across all scenarios
5. The compression result is weak. It is only done on easier environments in which communication is not quite necessary, especially 3s_vs_5z
6. An ablation on k is needed
7. The choice of baselines is not up to par. There needs to be more up-to-date baselines like MAPPO and IA2C
8.	It would be informative to see how the proposed techniques perform when added to existing communication methods not just QMIX and VDN.

[1] Tung, T. Y., Kobus, S., Roig, J. P., & Gündüz, D. (2021). Effective communications: A joint learning and communication framework for multi-agent reinforcement learning over noisy channels. IEEE Journal on Selected Areas in Communications, 39(8), 2590-2603.

[2] Lin, T., Huh, J., Stauffer, C., Lim, S. N., & Isola, P. (2021). Learning to ground multi-agent communication with autoencoders. Advances in Neural Information Processing Systems, 34, 15230-15242.

**Questions:**

1. Why is DCT only done in the last dimension? Seems arbitrary
2. What is the relationship between entropy and ‘ambiguity’? That is not well elaborated.

---

> ### Author Response · Authors · 2023-11-18
> **Rebuttal**
>
> Thank you for the helpful comments! We addressed them below but please let us know if you still have remaining questions.
>
> 1. We apologize for the imprecision in some of the language. The paper has been reviewed accordingly to address those. Regarding the specific points: with “messages with quality” we mean messages that allow the agents to solve the cooperative tasks thanks to communication. For instance, in most scenarios TARMAC performs poorly, meaning that the messages learned are likely to not be good enough. With an overload of the learning networks, we mean that very large and "heavy" messages can result in too large inputs, requiring very large networks that can be problematic in constrained systems.
> 2. Attention has been widely used due to its proven strength to improve communication in MARL. ARCOMM uses attention but the architecture is conceptually different from others. As described in section 3, we use embeddings of the observations as keys and values and message embeddings as values and, on top of that, we include a message regularizer during training that has shown to be very useful in learning, as proven by all the results in section 4. While we can see the link with noisy channels in the work mentioned by the reviewer, the overall approach is very different from our method, as well as the type of environments experimented. Yet, we included it in our related work, since it relates to lossy communication.
> 3. The autoencoder in the mentioned work indeed shares similarities to ours. However, they are conceptually different. The encoder in the autoncoder of AEComm takes as input the current observations to generate the following message and then the decoder tries to reconstruct the inputs. Instead, our regularizer in ARCOMM works as a recurrent encoder that takes the previous observations and messages and aims to predict the next observations. This is based on the premise that messages will help previous observations to predict the next observations. This work is however related to ours, and thus we have included it in our related work.
> 4. The results have variance due to the randomness of the tested environments. Still, we have run multiple seeds in order to minimise that variance and have tested in a range of different environments. We have not stated in the paper that ARCOMM is better in all scenarios. In fact, we have mentioned in section 4 that ARCOMM is not better in figure 4(c), PredatorPrey with maximum penalty, despite at the end of training it reaches the same optimal performance. The same happens in 3s_vs_5z where it does not start better than VDN but it ends the training at the same optimal level (additionally, we have noted that communication is not essential in this particular case). In all the other environments ARCOMM outperforms the others, meaning that it is more robust since it is consistently good across multiple scenarios while others are good in some of them, but completely fail in others. We have also carried out additional experiments to support our conclusions (revised **Appendix B**). As pointed by other reviewers (FWnu), communication in SMAC is not always necessary, but we highlight the improvements in the other tested environments like PredatorPrey (**Figure 4**) and the new Traffic Junction and Lumberjacks (**Appendix B**).
> 5. Compression is done not only in 3s_vs_5z but also in 2c_vs_64z (**Figures 5** and **10**), where we have shown that, despite communication not being utterly needed, it still has a strong positive effect. The compression result is naturally weaker since the agents are learning using lossy messages, i.e., messages that contain loss of information when compared to the original messages. Yet, as discussed in section 4, the point that we intend to make is that, even when using these lossy messages, the agents can still learn how to solve the tasks, despite the performance naturally dropping. This can be important in applications where communication is necessary but only lossy is possible, but the agents can still learn the tasks. Additionally, in **Appendix D**. we have shown the effect of compression in the messages learned.
> 6. During our experiments, we noticed that using high values of k can be detrimental since trying to predict many observations ahead can make the learning problem challenging. Thus, we opted to keep it as 1 in our final experiments.
> 7. We focus specifically on communication methods, and thus we believe that methods such as MAPPO or IA2C are not adequate for this paper, since they are not communication-based. We use VDN and QMIX which are not communication-based for the purpose of demonstrating the base performances without communication, and these have still shown very strong performances in these environments recently [1].
> 8. The proposed method is a communication method itself, for which we believe it would be unreasonable to apply it on top of other communication methods.

---

> > ### Author Response · Authors · 2023-11-18
> > **Rebuttal (Cont'd)**
> >
> > **Questions**
> > 1. DCT is done in the last dimension because this is the dimension of the vectors that contain the messages generated by each agent at each timestep to be sent to the others (section 2.4 and 3.2). Thus, we intend to compress the message. We apologise if this was not clear enough in the paper.
> > 2. Entropy evaluates the amount of uncertainty associated to the random outcome of a certain variable. Thus, if the entropy of the generated messages by the agents is higher, it means that there is more uncertainty associated with the messages, meaning that these will be more ambiguous. If, instead, the entropy is lower, it means that the messages are less uncertain and thus they are likely to be less ambiguous. This is shown in **Figure 6**, where we can see that messages with lower entropy are contained within smaller ranges, meaning that the values of the encoded observations can be represented within a smaller range of values, meaning that they are less ambiguous because the same base observations can be defined using less values.
> >
> > [1] Papoudakis, Georgios, et al. "Benchmarking multi-agent deep reinforcement learning algorithms in cooperative tasks." (2020).

---

> > > ### Comment · Reviewer_eV4L · 2023-11-21
> > >
> > > Thank you for your detailed response. Some of concerns have been addressed. Hence, I have increased my score from 3 to 5. But I still have reservations in the following aspects:
> > >
> > > 1. The novelty claim.
> > > 2. The high variance across results
> > > 3. The choice of baseline. I think non-communication baselines are important to add.

---

> > > > ### Author Response · Authors · 2023-11-22
> > > >
> > > > We would like to thank the reviewer for reading our responses and we are happy to have addressed some of the concerns!
> > > >
> > > > Regarding the remaining aspects we would like to add the following:
> > > >
> > > > 1. Our novelty lies mainly in the architecture proposed for communication that uses a conceptually different attention module where queries are message embeddings and keys and values are embeddings of the observations. This attention module is supported by our message regularizer which is also novel and has been shown to be beneficial for learning. Additionally, we analyse lossy communication in MARL from a different perspective using the DCT, which is widely used in practical applications.
> > > >
> > > > 2. This high variance is due to the seeds and complexity of the environments used. We are currently running more seeds in order to minimise this variance. We would also like to emphasize that, despite the variance, the proposed method is robust across environments, while the others perform well in some of them but poorly in others.
> > > >
> > > > 3. We do use non-communication methods as baseline such as VDN and QMIX, that have shown also in other works to be strong in these environments. Nevertheless, we are currently running other additional non-communication methods to complement our experiments.

---

### Official Review · Reviewer_FWnu · 2023-10-31

**Soundness:** 2 fair
**Presentation:** 2 fair
**Contribution:** 2 fair
**Rating:** 5
**Confidence:** 4

**Summary:**

The paper studies Multi-agent communication in complex scenarios with limited communication channels. The authors introduces Attentive Regularized Communication (ARCOMM), a method for communication in Multi-Agent Reinforcement Learning (MARL) that uses an attention module and a message regularizer to improve the performance of agents in complex cooperative tasks. Additionally, the paper investigates the use of the Discrete Cosine Transform (DCT) for compressing messages into a smaller representation, without affecting the size of communication endpoints. The results show that ARCOMM improves agent performance compared to other methods in SMAC environment and supports the motivations for the proposed method.

**Strengths:**

1.	Investigating lossy information-based compression through the Discrete Cosine Transform (DCT) to reduce message sizes, and analyzing the effectiveness of this compression scheme in scenarios with limited communication channels.
2.	Showing that agents can still learn even with lossy communication, and that the messages learned by ARCOMM support the motivations for the proposed method.

**Weaknesses:**

1.	Incremental novelty w.r.t TarMAC by only adding a message regularizer.
2.	The paper only evaluates the performance of ARCOMM in scenarios with lossy communication channels using the Discrete Cosine Transform (DCT) for message compression. It does not explore other compression techniques or investigate the impact of different levels of compression on agent performance.
3.	Only SMAC is used as a testbed, where parameter sharing is more important than communication.

**Questions:**

1.	The paper does not provide a comprehensive analysis of the computational complexity or scalability of ARCOMM. It would be valuable to understand how the method performs in larger-scale MARL scenarios or with a higher number of agents.
2.	Maybe the authors can drop the parameter sharing in SMAC in the ablation study.
3.	Could the authors provide some example about how to use DCT to compress the message vectors?

---

> ### Author Response · Authors · 2023-11-18
> **Rebuttal**
>
> Thank you for providing useful comments to our paper! We are happy that the reviewer agrees with the value of studying lossy communication. We have addressed your comments below but please do let us know if there are still any remaining concerns.
>
> **Relation with TARMAC**
>
> ARCOMM is structurally different from TARMAC, despite both using attention modules and the message regularizer is not the only difference. As described in section 3, ARCOMM uses embeddings of the observations and keys and value and message embeddings as queries, following the motivations described in this section. Additionally, the message regularizer proves to be a strong add-on to the network architecture, as depicted in section 4. These factors compose big structural differences that drastically improve the performance of our method, as shown in the paper. Additionally, we have added extra experiments in the revised **Appendix B**.
>
> **Compression**
>
> One of the contributions of this work is to show how the DCT can be used in the context of MARL. The DCT is widely used in realistic applications regarding information compression, and thus it can be an important factor to include within MARL approaches, in order to approximate them to more realistic applications. However, we do intend to include other compression methods in the future. Regarding the analysis of different levels of compression, we indeed do analyse the performance of different levels of compression with the DCT in **Figure 5** and in **Appendix D: Figure 10**.
>
> **Tested Environments**
>
> It is not true that only SMAC is used as a testbed. In **Figure 4** we show the performance of the proposed method in a range of PredatorPrey tasks with different penalties. While in SMAC communication might not be necessary (as pointed by the reviewer), in these other tasks that we have tested communication is crucial. This shows that our method can perform well in cases like SMAC as well as in cases where communication is crucial, while the other methods might perform well in some cases but do not hold for others. Nonetheless, we have run additional experiments and show results in more environments where communication might be more beneficial (Lumberjacks and Traffic Junction with 10 agents) in the revised **Appendix B**.
>
> **More agents**
>
> We have tested the proposed method in MMM2, an environment with 10 agents, which is one of the environments with more agents in SMAC. Thus, we believe that showing robustness with 10 agents can give a good idea about the scalability of ARCOMM. In addition to this, we have also done additional experiments where we tested in a traffic junction environment with 10 agents with strong penalties, posing a very complex scenario (**Appendix B**).
>
> **Parameter Sharing**
>
> We are very interested in the study of dropping parameter sharing in the type of scenarios analysed. However, that will need more complex considerations and it is out of the scope of this paper. However, this is something that we intend to do in the near future.
>
> **Compression with DCT**
>
> We have used a pytorch DCT module to compress the messages of the agents. More specifically, as it is mentioned in sections 2.4 and 3.2, after the agents generate their messages, we pass them through the DCT function that is applied on the last dimension of the vectors, i.e., the messages corresponding to each agent at each timestep. These compressed messages are then sent to the others, where the inverse is applied and the messages are decompressed, incurring into some loss of information.

---

> > ### Comment · Reviewer_FWnu · 2023-11-21
> >
> > I read the response and decided to keep my score.

---

> > > ### Author Response · Authors · 2023-11-22
> > > **Thanks**
> > >
> > > Thank you for taking the time to read our response! We hope to have addressed the initial comments with the responses and changes to the manuscript. Are there still any remaining concerns that we can further address in order for the reviewer to raise the score?

---

> > > > ### Comment · Reviewer_FWnu · 2023-11-22
> > > >
> > > > Authors solve the part of my concerns by adding new experiments in Appendix B. However, there is still a concern about novelty as pointed by other reviewers.

---

> ### Author Response · Authors · 2023-11-22
>
> Thanks for the clarification, we are happy that part of the concerns were addressed thanks to the new Appendix B!
>
> With regards to the remaining question, we would like to add that our novelty lies mainly in the architecture proposed for communication that uses a conceptually different attention module where queries are message embeddings and keys and values are embeddings of the observations. This attention module is supported by our message regularizer which is also novel and has been shown to be beneficial for learning. This is all very different from other methods such as TARMAC, for example (as mentioned by the reviewer), as supported by the much-improved performances of the proposed method. In addition to this, we analyse lossy communication in MARL from a different perspective using the DCT, which is widely used in practical applications.

---

### Official Review · Reviewer_cGtd · 2023-10-31

**Soundness:** 2 fair
**Presentation:** 3 good
**Contribution:** 2 fair
**Rating:** 3
**Confidence:** 4

**Summary:**

This paper introduces Attentive Regularized Communication (ARCOMM), a method for communication in MARL. ARCOMM incorporates an attention module and a message regularizer to assess the significance of messages and facilitate the learning of more meaningful interactions among agents. The study suggests that ARCOMM is robust, maintaining learning capabilities even under conditions of limited or lossy communication channels.

**Strengths:**

- the paper is relatively easy to follow
-

**Weaknesses:**

- The novelty of the paper seems limited. The performance of the proposed method is not significantly better than QMIX based on Figure 2. It is also surprising to see that the baselines with network communication have lower performance than the ones without communication.
- The related works discussed in the paper are not complete for communication-based MARL. Techniques such I2C[1], VBC[2], TMC[3], and many more are not discussed. They need to be compared as well for your experiment.


[1] Ding, Ziluo, Tiejun Huang, and Zongqing Lu. "Learning individually inferred communication for multi-agent cooperation." Advances in Neural Information Processing Systems 33 (2020): 22069-22079.
[2] Zhang, Sai Qian, Qi Zhang, and Jieyu Lin. "Efficient communication in multi-agent reinforcement learning via variance based control." Advances in Neural Information Processing Systems 32 (2019).
[3] Zhang, Sai Qian, Qi Zhang, and Jieyu Lin. "Succinct and robust multi-agent communication with temporal message control." Advances in Neural Information Processing Systems 33 (2020): 17271-17282.

**Questions:**

- How is the proposed method able to achieve better performance compared to TarMAC, which is also an attention-based communication technique? More ablation is required to help better understand which part of the proposed design really improves the performance.

- Figure 3 compares QMIX and QMIX + ARCOMM. How does QMIX + ARCOMM compared to other communication-based methods?

---

> ### Author Response · Authors · 2023-11-18
> **Rebuttal**
>
> Thanks for taking the time to review our paper and for your helpful comments! We address them below and please let us know if there are still any remaining concerns.
>
> **Novelty**
>
> Performances are not significantly better in Figure 2 since SMAC does not necessarily require communication, as discussed in our results and also pointed by other reviewers (FWnu). Yet, the proposed method still shows improvements mainly in the more complex cases, MMM2 and 2c_vs_64zg. However, when we look at environments where communication is needed, as in the case in **Figure 4**, we can see that our method clearly performs better than the others. Importantly, this shows that our method can perform well across environments, while others will perform well in some of them but poorly in others. Additionally, we have run extra experiments in scenarios where communication might have a stronger impact in order to highlight the strength of the proposed method (**Appendix B** of the revised paper).
>
> The baselines with network communication have lower performance in some scenarios since these communication architectures might not be robust enough in some cases, as shown in other works such as [1]. ARCOMM on the other hand provides a more efficient architecture that can perform well in all tested scenarios.
>
> **Benchmarks and Related Work**
>
> We opted to place the Related Work section in Appendix A as a structural option, we apologize if that went unnoticed. In that section we discuss several communication-based methods for MARL. Additionally, we also discuss part of them in the introduction. We appreciate the methods pointed by the reviewer, that we have added to our related work. However, after going through the codes of these methods, we noticed that the code of I2C is not prepared to run on environments such as SMAC (that is not tested in their paper) and VBC and TMC are hardcoded to run in just a small selection of SMAC environments. In our work, we test with a variety of environments that are not only from SMAC (where communication is not always required). Thus, we cannot use these methods as benchmark since they cannot be directed applied out of the box in other environments. On the other hand, our method is completely generalisable and can run on the go in any environment.
>
> **Relation with TARMAC**
>
> ARCOMM is conceptually different from TARMAC. Despite both of them using attention to learn messages, as described in section 3 ARCOMM uses embeddings of the observations for keys and values and message embeddings for queries. On top of this detail, ARCOMM introduces a message regularizer that proves to be very useful for the learning process. The superior performance of ARCOMM is mainly due to these two structural factors. On top of that, TARMAC has shown in other works to not be very robust in these scenarios. On the other hand, ARCOMM shows robustness and performs well across the range of tested environments. We have run additional experiments to further validate our method and we have tested QMIX+ARCOMM without the regularizer. With this being said, the proposed method is very conceptually different from TARMAC and thus is capable of demonstrating much better performance.
>
> **Figure 3**
>
> Figure 3 was used for the purpose of evaluating the effect of communication when comparing to not using communication. We opted to not include TARMAC as it performs poorly in this environment. However, we show in **Appendix C** of the revised paper the behaviour also for QMIX+COMMNET.
>
> [1] Liu, Z., Wan, L., Sun, K., & Lan, X. "Multi-Agent Intention Sharing via Leader-Follower Forest". AAAI (2021).

---

> > ### Author Response · Authors · 2023-11-22
> >
> > We thank the reviewer once again for taking the time to review our paper! We hope to have addressed your concerns with our responses and revisions in the manuscript. As the rebuttal period is reaching an end, we would like to ask if there are still any further questions that we can address in order for the reviewer to raise the score.

---

### Official Review · Reviewer_j1DY · 2023-11-01

**Soundness:** 1 poor
**Presentation:** 2 fair
**Contribution:** 2 fair
**Rating:** 3
**Confidence:** 5

**Summary:**

This paper focuses on reducing communication overhead in communication MARL. It utilized an attention module and DCT to compress the messages.

**Strengths:**

1. Communication overhead is an important direction in communication-based MARL.

**Weaknesses:**

1. The contribution is rather limited. Tarmac is the first paper to utilize attention modules. Many papers have followed its architecture since then. In addition, many papers also investigated the communication overhead, such as I2C.

2. The experiment part is weak. More communication-based methods and more SMAC maps (at least 6-8) should be included for a more comprehensive comparison  In addition, at the current stage, the improvement is not significant. It is hard to tell whether the method is effective.

3. For Figure 3, showing the difference between the communication-based method and the non-communication method is not meaningful since many papers have proved this. Many interesting behaviors against other communication-based methods should be excavated.

4. Related work is missing. Apparently, many works have adopted attention modules to process redundant information as a standard procedure. I encourage the authors to fully investigate the communication-based MARL.

**Questions:**

For the compression part, are any learned parameters included?

---

> ### Author Response · Authors · 2023-11-18
> **Rebuttal**
>
> Thank you for your useful feedback! We have addressed below your comments. We hope that we have clarified your concerns, but please do let us know if there are still questions remaining.
>
> **Contribution and relation with TARMAC**
>
> It is true that other papers have investigated attention since it is a compelling way to learn useful messages for multi-agent communication. For that reason, multiple works have focused lately in this kind of architectures for communication. Our proposed architecture is conceptually different from others and, in particular, from TARMAC. ARCOMM uses an attention module where the keys and values are embeddings of the observations and queries are message embeddings. Additionally, this attention module is supported by a message regularizer that we have shown to be beneficial. This is very different from other architectures, and from TARMAC in specific.
>
> Regarding communication overhead, we do not focus on communication overhead in this paper. We focus on investigating whether agents can still solve tasks using communication even when the communication channels are lossy, i.e., have limited channel capacity, whereas communication overhead corresponds to the communication rate among agents. To evaluate lossy communication we used the DCT. This is very different from communication overhead being analysed in I2C.
>
> **Experiments**
>
> We have experimented with SMAC methods to showcase that our method is robust across different environments. The improvements are not significant in 3s_vs_5z since this environment like others in SMAC can be solved without communication, as pointed also by other reviewers (FWnu). Still, we opted to include SMAC maps to show that our method is robust across environments, and show how it still improves performances in more complex cases such as MMM2 and 2c_vs_64zg. In **Figure 4**, we show how it outperforms the others in **PredatorPrey** tasks with different penalties (this type of environment have been used in multiple other communication works). Yet, to support our method, we have conducted more experiments in an additional complex SMAC map (3s5z) and in two more environments where communication is important due to their complexities – Traffic Junction with 10 agents and Lumberjacks. Our method also performs well in these cases (**Appendix B** of the revised paper).
>
> Regarding Figure 3, we have analysed these behaviours in order to highlight the improvements that the proposed communication method brings in terms of agent behaviour. To make this analysis more meaningful we have included an additional behaviour for QMIX+COMMNET is **Appendix C**.
>
> **Related Work**
> We opted to place the related work section in Appendix A due to a structural choice. We have in fact discussed multiple of these communication methods in the introduction and related work (in Appendix A) sections. We apologize if that went unnoticed.
>
> **Questions**
> The DCT is a deterministic compression algorithm widely used in information theory and signal applications, so no parameters are learnable in the compression procedure.

---

> > ### Author Response · Authors · 2023-11-22
> >
> > Thanks once again for the time taken to review our paper! We hope to have addressed your concerns in our responses and with the revisions made to the manuscript. As the rebuttal period is coming to an end, we would like to ask if there are any further questions that we can address in order for the reviewer to raise the score.

---

### Author Response · Authors · 2023-11-18
**General Response**

We would like to thank all the reviewers for their time and feedback given to improve our paper. We have updated a revised version with additional experiments (**Appendix B**) and some other clarifications. All the changes are marked in red.

We have addressed the questions of the reviewers individually. However, we would like to reiterate some points that were mentioned by most reviewers:
* **Relation with TARMAC**: Despite both using attention mechanisms, ARCOMM is conceptually different from TARMAC. As described in section 3, ARCOMM uses embeddings of the observations as keys and values and messages embeddings as queries. Adding to that mechanism, ARCOMM uses a message regularizer that has been shown in Figures 2, 4, 6 and 8 to be very beneficial for the message generation process and overall team performance. Thus, ARCOMM's is conceptually very different from TARMAC. As a result, we note the general improvements of performance as demonstrated in this paper, whereas TARMAC performs poorly.
* **Benchmarks and related work**: We have presented a discussion of multiple related methods in the related work section in Appendix A. We placed it in the appendix as a structural choice but we apologise if it went unnoticeable. Unfortunately, some of these related methods did not open-source their codes. The ones mentioned by reviewer cGtd are indeed useful to include in the related work, but their official implementation seems to be hardcoded for certain environments that are not the ones tested here. On the other hand, our method is generalisable and works for any environment out of the box. In fact, we had to implement ourselves some of the other baselines, apart from TARMAC.
* **ARCOMM’s performance**: The reviewers have raised concerns regarding the performance of the proposed method. ARCOMM (either VDN+ARCOMM or QMIX+ARCOMM) performs better than all the others in all the environments except PredatorPrey with maximum penalty (figure 4(c)) and 3s_vs_5z. However, in both these cases, ARCOMM ends training at the same level as the other best method (VDN in 3s_vs_5z and VDN+COMMNET in figure 4(c)). Additionally, in 3s_vs_5z we have noted that communication is not essential. Overall, ARCOMM performs well in every task attempted and outperforms the others aside from these two cases. This means that ARCOMM is more robust and can perform well in the wide range of environments attempted, while the others might perform well in some cases, but completely fail in others (as discussed in section 4). Our additional experiments in **Appendix B** also support this.

---

### Meta-Review · Area_Chair_puBn · 2023-12-11

**Metareview:**

This paper proposes ARCOMM, a method for communication in multi-agent reinforcement learning (MARL). ARCOMM incorporates an attention module to evaluate message weights and a message regularizer to enhance the meaningfulness of messages, leading to improved team performance. The study also analyzes how ARCOMM performs in scenarios where message compression is necessary and demonstrates its effectiveness in enhancing agent performances compared to other methods in complex domains through the power of communication. There are some weaknesses or concerns of the paper raised from the review comments and discussions, including the incremental technical novelty (the relation between ARCOMM and TarMAC), discussion of related works, experiment sufficiency. Although the authors provided detailed feedbacks, some of the concerns raised are still unsolved.

**Justification For Why Not Higher Score:**

There are some weaknesses or concerns of the paper raised from the review comments and discussions, including the incremental technical novelty (the relation between ARCOMM and TarMAC), discussion of related works, experiment sufficiency. Although the authors provided detailed feedbacks, some of the concerns raised are still unsolved.

**Justification For Why Not Lower Score:**

N/A

---

### Decision · Program_Chairs · 2024-01-16

Reject